# Helmet Shape and Phylogeography of the Treehopper *Membracis mexicana*

**DOI:** 10.3390/insects14080704

**Published:** 2023-08-14

**Authors:** Marisol De-la-Mora, Daniel Pinero

**Affiliations:** 1Escuela Nacional de Estudios Superiores Campus Juriquilla, Universidad Nacional Autónoma de México, Boulevard Villas del Mesón 3001, Querétaro 76230, Mexico; 2Departamento de Ecología Evolutiva, Instituto de Ecología, Universidad Nacional Autónoma de México, Ciudad Universitaria, Ciudad de México 04510, Mexico; pinero@ecologia.unam.mx

**Keywords:** treehopper, Membracidae, Mexico, biogeographic provinces, phylogeography

## Abstract

**Simple Summary:**

*Membracis mexicana* is a treehopper widely distributed in the neotropical region where it is a minor pest of some crops. In Mexico, it is found in at least four biogeographic provinces (abiotic subdivision of biogeographic realms where assemblages of biota share an evolutionary history), and few species of insects have such wide distributions. Field observations indicate that there are different forms of this species, but so far, how the different forms appear and how they are distributed have not been recorded. To test whether the distribution of morphological and genetic variation was consistent with the regionalization of the biogeographic provinces, we analyzed the morphological and genetic variation of this species sampling a total of 303 insects and conducted a geometric morphometric analysis from the body shape and phylogeographical analysis using five sequenced genes. We found three different morphotypes of these species and two genetic groups; this means that one of the genetic groups shows two forms. These findings show that *M. mexicana* is a very variable species geographically structured by the conformation of biogeographic provinces.

**Abstract:**

*Membracis mexicana* (Hemiptera: Membracidae) is distributed in four biogeographic provinces of Mexico. Field observations indicate that there are different forms of this species, but the distribution of the phenotype and the genetic variation of this species have not been clarified. The aim of this study was to quantify the phenotypic and genetic variation of *M. mexicana* and determine whether the configuration of biogeographic provinces impacts the distribution of this variation. To achieve this, we analyzed 307 photographs using 19 landmarks and geometric morphometrics to quantify the phenotypic variation in helmets. We sequenced five molecular markers for 205 individuals to describe the phylogeographic pattern. As a result, we identified three morphological configurations of the helmet of *M. mexicana* and two genetic lineages. The morphotypes are (1) a large and wide helmet with small dorsal spots, (2) a small and narrow helmet with large dorsal spots, and (3) a small and narrow helmet with small spots. Genetic lineages are distributed in southeast and western Mexico. The western lineage corresponds to two helmet morphotypes (1 and 2) and the southeast lineage to morphotype 3. We found that the larger helmets correspond to the western lineage and are distributed in Trans-Mexican Volcanic Belt and Pacific lowlands provinces, whereas the smallest helmets correspond to the southeast lineage and are present in the Veracruzan and Yucatan Peninsula provinces.

## 1. Introduction

The biogeographic provinces of Mexico were formed during the Cenozoic, and their current configuration is due to recent events such as volcanic activity starting at the Pliocene 5.33 million years ago (Ma) and continuing to the present [1], the movement of tectonic plates that closed the Isthmus of Panama (3 Ma) [2] and dropped the region’s temperature between 7 and 5 °C during the maximum glaciation, and the glacial cycles that promoted altitudinal movements of the biota during the Pleistocene 2.5–0.011 Ma [3]. These events generated a complex transition zone in Mexico, with consequences on soil formation, geological changes, and different precipitation ranges, consequently causing a complex assemblage of biota between Nearctic and neotropical affinities as well as in situ speciation [4,5]. In this scenario, plant species evolved [6] and habitats for phytophagous insects were generated.

According to the characterization of Morrone et al. [7], 14 biogeographic provinces in Mexico are recognizable because they combine climatic, geological, and biotic criteria. The biogeographic provinces represent historical entities of specific interactions of biota. The Trans-Mexican Volcanic Belt (TMVB) province has documented a higher diversity of ecosystems and species compared with other biogeographic provinces [7,8,9]. If there are more biotic interactions at the TMVB, one might expect there to be repercussions increasing the variation (phenotypic and genetic) in this area of a widely distributed species. 

Here, we investigate the intraspecific variation of the treehopper species Membracis mexicana distributed over four biogeographic provinces: the Pacific Coast, the TMVB, the Mexican Gulf, and the Yucatan Peninsula. We aimed to demonstrate phenotypic variation and phylogeographic patterns supported by the biogeographic provinces, hoping to find more diversity at the TMVB compared with other biogeographic provinces.

The treehopper species (family Membracidae) are characterized by the shape of the helmet (complex three-dimensional structures that develop from the pronotum and cover the dorsal side of the insect). Species of the genus Membracis show intraspecific variation in the size and colors of the helmet; this variation has been attributed to the host plant, nutrition, temperature, and humidity [10], but until now, these hypotheses have not been tested. *M. mexicana* is by far the most abundant species of the genus in Central America [11] and is also very variable in size and color (Figure 1). It is an excellent model species to analyze intraspecific variation at the phenotypic and genetic levels. In this sense, we analyzed the helmet shape to characterize the phenotype and used five markers: four nuclear sequences (genes: 28S ribosomal, H2A, H3, and Wg) and one mitochondrial (COI) to estimate the genetic diversity of *M. mexicana.*

## 2. Materials and Methods

### 2.1. Study Area

In Mexico, *Membracis mexicana* is distributed throughout the tropical region along both the Pacific and Gulf coasts, including the Yucatan Peninsula. Collection permits were issued by SEMARNAT, official letters No. SGPA/DGVS/11707/19 and SGPA/DGVS/06171/20. It is important to point out that this species is not in any risk category according to NOM-059 of SEMARNAT. Based on the records of insects collected and deposited in the National Insect Collection of the IB, UNAM, 15 points along the two coasts of Mexico and the center were visited. Four sampling seasons were organized in May 2019 (visiting the states of Veracruz, Puebla), December 2019 (states: Veracruz, Campeche, Quintana Roo, and Yucatán) December 2021 (states: Michoacán, Guerrero), and May 2021 (states: Michoacán, Jalisco, Nayarit, Guadalajara, Colima). At each collection site, we looked for the host plants reported in the literature and reviewed the plants that predominated in the area. The points were carefully chosen to represent the distribution of *M. mexicana* in Mexico (Table 1). Additionally, sequence samples from the GenBank of *M. mexicana* were added to the south of its distribution, in Guatemala and Panama (COI: AY513459.1 and AY513460.1; Wg: AY593701.1 and AY593700.1) and other species of Membracis for phylogenetic analysis. Species: *M. luizae* (COI: KX924959.1; Wg: KX925187.1; H3 KX925047.1; 28S: KX924867.1; H2A: KX924705.1)*, M. foliata* (COI: KF919641.1; Wg: AY593697.1), *M. flava* (COI: AY513455.1; Wg: AY593696.1), and *M. trimaculata* (COI: AY513457.1; Wg: AY593698.1).

### 2.2. Insect Sampling 

We collected between 20 and 30 adult individuals per population (Table 1). Each one was put in a tube with absolute alcohol and its respective label for subsequent DNA extraction. All host plants were recorded. The upper and lower sides of small plants and shrubs were checked to manually collect the specimens from different leaves or plants with the aim of not taking genetically related individuals. It has been reported that the adults move very short distances, and the juveniles remain feeding on the same leaf until their fifth molt, while the mother takes care of them [10]. 

### 2.3. Geometric Morphometrics

We took pictures of each sampled insect in RAW format with a Canon 600 d camera equipped with an EF-S 60 mm macro lens. F2.8 (Canon (UK) Ltd., Surrey, UK). Each image included a ruler of 1 mm as background and a standard color chip (Colorgauge Micro, Image Science Associates LLC, Williamson, NY, USA). The images were processed with Adobe Photoshop Lightroom software (version 24.1.0, Adobe Inc., San José, CA, USA) (Adobe Systems Software Ireland Ltd. 2023, San José, CA, USA).

Two-dimensional landmarks and the scale factor were recorded using the tpsDig software (version 2.33, SB Morphometrics, NY, USA) [12]. We used nineteen landmarks to capture the shape of the helmet (Appendix A). The landmark coordinates of all specimens were aligned using a Procrustes superimposition, and shape information was extracted from the landmark data. This procedure eliminates the variation of landmark configurations due to size differences, position, and orientation. To compute the centroid size, we used the tpsRelw software (v1.75, SB Morphometric, NY, USA) [13] and its size was compared between populations with an ANOVA. The measure of the helmet shape differences between two forms, the Procrustes distance, is the square root of the sum of squared distances between the landmarks after Procrustes superimposition [14], estimated using MorphoJ (v1.08.0, Klingenberg lab, MC, UK) [15]. We conducted a principal component analysis (PCA) to explore shape data with any restriction in sample size or clustering variable. PCA produces a new set of variables (the principal components, PCs) that are not correlated [15,16]. To compare the helmet shape between populations or host plants, we implemented a canonical variate analysis, a widely used method for analyzing group structure in multivariate data; this was performed with MorphoJ [15]. To visualize the shape differences between the mean landmark configuration and the configuration shifted up or down along the respective PC axis of the PCA and CV axis of the CVA, we chose the lollipop diagrams using MorphoJ and edited the plots in Adobe Illustrator (v27.8, Adobe Inc., San José, CA, USA).

### 2.4. DNA Extraction, Sequencing, and Processing

The DNA extraction of the sampled insects was made with the Qiagen^®^ DNeasy kit for blood and tissue. Except for the pronotum, the complete body of the insect was used for extraction. It was placed in a 1.5 ml tube for processing, liquid nitrogen was used to homogenize the sample using a micropistil, and then the protocol of the kit was followed. The amount of DNA extracted was quantified with the NanoDrop ND1000 (Thermo Fisher Scientific Inc. Waltham, MA, USA). For the amplification of the five gene regions (28S, H2A, H3, Wg, and COI), we used the oligos and conditions mentioned in Evangelista et al. [17]. The chromatograms were analyzed and the sequences were edited with the Chromas (v2.6.6, Technelysium DNA Sequencing Software, QLD, Australia). For each sample, the quality of the sequences was checked and the ends where the bases had a quality of 30 or less were eliminated. It was visually confirmed that the peaks of the chromatogram corresponded with the read base.

The identity of the obtained sequences was verified via BLAST on the National Center for Biotechnology Information portal (http://blast.ncbi.nlm.nih.gov/Blast.cgi Accessed on 16 June 2022) using the blastn optimization algorithm to identify and remove contaminated sequences. For each database of the sequence of each gene, the alignment was made with Clustal Omega (v 2.0.12, EMBL-EBI, Europe) [18]. The evolution model of each sequence was estimated with the jmodelTest program (v2.1.10, Posada Lab., Pontevedra, Spain) [19].

### 2.5. Genetic and Phylogeographic Analyses

The measures of polymorphism and genetic diversity (the number of haplotypes, the values of haplotypic diversity, nucleotide diversity, and theta value) and the neutrality test of each gene were estimated with the software DnaSP (v 6, University of Barcelona, Barcelona, Spain) [20]. We used SAMOVA (v 2.0, University of Bern, Bern, Switzerland) [21] running from K = 2 to K = 11 (number of sampled sites) for each gene sequence to determine the population structure. Haplotype networks for each gene were constructed with the median-joining algorithm [22] in Network (v.2.0, Fluxus Technology Ltd., Cambridge, UK) and PopArt (v1.7, University of Otago, Dunedin, New Zealand) [23].

To infer the genealogical lineages, we estimated the phylogeny per gene by Bayesian inference (BI) with BEAST2 (v 2.7.5, Centre for Computational Evolution, University of Auckland, Auckland, New Zealand) [24]. We used the COI and Wg gene sequences from GenBank to root the phylogeny. BEUTi (Centre for Computational Evolution, University of Auckland, Auckland, New Zealand) was used to prepare the input, Tracer (v.1.7.1, Centre for Computational Evolution, University of Auckland, Auckland, New Zealand) [25] to eliminate burn-in and examine the convergence of chains, TreeAnotator (Centre for Computational Evolution, University of Auckland, Auckland, New Zealand) to summarize the obtained trees, and FigTree (v.1.4.4, Molecular Evolution, Phylogenetics and Epidemiology, The University of Edinburgh, Edinburgh, UK) for its visualization. The running parameters were 10,000,000 generations for each gene, and 1000 generations of burn-in were removed. We also estimated a single phylogeny using the four nuclear sequences using a concatenated matrix and *M. luizae* as an outgroup by BI using MrBayes (v3.2, Department of Biodiversity Informatics, Swedish Museum of Natural History, Stockholm, Sweden) running 10,000,000 generations, until the sampling variance reached 0.01.

For the analysis of demographic reconstruction, we used the mutation rates reported in Allio et al. [26]: mmit = 0.009 mutations per site per million years for the mitochondrial sequence and mnuc = 0.0044 mutations per site per million years for the data set with four nuclear sequences. Both were run under the Coalescent model, a Coalescent Bayesian Skyline analysis in BEAST v.2 [24]. The analyses were run for 30,000,000 generations and 3000 generations of burn-in were removed. For the convergence of chains and the visualization of Skyline Plots, we used Tracer v.1.7.1 [25]. 

## 3. Results

### 3.1. Insect Sampling 

Of a total of 15 sites visited, we found *M. mexicana* in 11 (Table 1). We observed adults and eggs in six plants and only adults in *Heimia salicifolia.* In total, 309 adults of *M. mexicana* were sampled on seven plant species: *Terminalia catappa, Ficus carica, Spondias purpurea*, *Artocarpus altilis, Cassia fistula, Heimia salicifolia*, and *Salix babylonica.* In additional observations, we sampled the Pacific coast of Mexico in December 2020, but did not find *M. mexicana* on any host plant.

### 3.2. Geometric Morphometric Analysis 

#### 3.2.1. Centroid Size of the Helmet

The centroid size by biogeographic province showed different sizes (Figure 2A), indicating that the Pacific lowlands and TMVB provinces have the largest helmet compared to the other provinces. The difference is statistically significant (F = 42.45, *p* = < 2 × 10^−16^). The distribution of the data indicates that the helmet of these populations range from 9 to 14 units, while those of the other populations are from 7 to 11 units.

The centroid size by the host plant (Appendix A) shows that the helmets sampled on *T. catappa* are of all sizes, but the largest helmets were sampled on *Spondias purpurea, Artocarpus altilis, Cassia fistula, and Heimia salicifolia*, and the smallest on *Ficus carica* and *Salix babylonica* (F = 7.894, *p* = 6.58 × 10^−0.8^).

#### 3.2.2. Principal Components of Helmet Shape

The analysis of principal components (Appendix A) shows that the principal axes that describe the shape of the helmet are the width with 24% (principal component 1) and the length with 20% (principal component 2). PC 1, the width of the helmet, separates the Pacific lowlands and the TMVB (Colima, Puerto Vallarta, San Patricio, Santa Teresa, and Tepic) from the other provinces, except for the Zacapu population at the TMVB, whose helmet shape remains in the center of the distribution. The length of the helmet (PC2) does not allow us to discern groups; the insects in the populations seem to be distributed homogeneously, with long and short pronotums in all populations. The analysis of canonical variance by the host plant takes away the insects sampled on *Salix babylonica* from the rest, also from the Zacapu population (Appendix A). The canonical variate 1 (CV1) shows that the size of the dorsal spots is largest on their helmet and is what makes this population different from the others.

#### 3.2.3. Canonical Variance of Helmet Shape

The analysis of canonical variance by biogeographic province indicates three groups (Figure 2B). In group 1, we have the Pacific lowlands and TMVB provinces, in group 2, only the TMVB (Zacapu), and in group 3, the Veracruzan and Yucatan peninsula provinces. The canonical variate 1 (CV1) is related to the width of the helmet: the widest helmet occurs in group 1, the narrowest in group 3, and the intermediate ones in group 2. The canonical variate 2 (CV2) has mainly to do with the size of the dorsal spots. These only differentiate the population of Zacapu (in the TMVB) from the rest, indicating larger spots.

Altogether, we found phenotypic variation in three components of the helmet of *Membracis mexicana*: size, width, and size of the dorsal spots. The interpretation of (large and small) helmet size is based on centroid size, a measure that is estimated from the landmarks on the periphery of the figure towards the central point. The interpretation of width or length is achieved using principal component analysis, in which the helmet deforms according to the variance components, expanding up and down what we consider to be the width, and backward and forward what we consider to be the length. In the particular case of the size of the dorsal spots, it is the distance between the landmarks 4–5 and 6–7 (see Appendix A). This variation is not randomly distributed. We found helmet type 1 (large, wide, and with small dorsal spots) in Tepic, Colima, Puerto Vallarta, San Patricio, and Santa Teresa. Helmet type 2 (small, narrow, and with large dorsal spots) was found in Zacapu, and helmet type 3 (small, narrow, and with small spots) in Veracruz, Coatzacoalcos, Ciudad del Carmen, Champoton, and Cancun. This variation is distributed in the biogeographical provinces as follows: helmet type 1 in the Pacific Coast (Puerto Vallarta and San Patricio), helmet types 1 and 2 in the TMVB province (Tepic, Colima, Santa Teresa, and Zacapu), and helmet type 3 in the provinces of the Gulf of Mexico (Veracruz and Coatzacoalcos) and the Yucatan Peninsula (Ciudad del Carmen, Champoton, and Cancun).

### 3.3. Genetic and Phylogeographic Analyses

#### 3.3.1. DNA Polymorphism 

A total of 844 sequences were obtained: 189 for the 28S gene, 177 for the H3 gene, 190 for the H2A gene, 147 for the Wg gene, and 141 for the COI gene (Appendix A). The haplotype sequences are deposited at GenBank (ID numbers are: OR123974-OR124006 (for H2A), OR124007-OR124028 (for H3), OR124029-OR124058 (for Wg), OR120258-OR120285 (for 28S), and OR120103-OR120156 (for COI)). The diversity indices indicate high genetic diversity in all genes. For example, the haplotype diversities were 0.797 (28S), 0.625 (H3), 0.425 (H2A), 0.874 (Wg), and 0.881 (COI), while the nucleotide diversities were 0.01071 (28S), 0.00342 (H3), 0.00375 (H2A), 0.03121 (Wg), and 0.01807 (COI). The neutrality tests for all genes were statistically significant: the Fu and Li’s F −6.41793 (28S), −3.86732 (H3), −4.04503 (H2A), and −2.85538 (COI), and only the Wg gene was positive (2.36933). Tajima Ds were significant and negative for 28S, H3, and H2A (−2.18358, −1.97102, and −2.34419), while they were positive and significant for the Wg gene (2.81039) and negative and non-significant for COI −1.72698). Figure 3 shows the haplotype diversity of each gene by biogeographic province. 

#### 3.3.2. Haplotype Networks 

The haplotype networks for all four genes (Figure 4; Appendix A) and their geographic distribution (Figure 4; Appendix A) show large divergence and clear geographic structure. The numbers of each haplotype and the frequency of haplotypes are shown in the Appendix A. For all nuclear genes, the most frequent haplotypes are found in all populations. Finally, COI amplification was not successful in the samples from Puerto Vallarta, San Patricio, and Colima.

#### 3.3.3. Analysis of Population Structure 

The pattern of the population structure shown by the five genes indicates a genetic break between the western populations (Tepic, Puerto Vallarta, San Patricio, Colima, Santa Teresa, and Zacapu) and the southeastern populations (Veracruz, Coatzacoalcos, Ciudad del Carmen, Champoton, and Cancun). The geographical distribution of each of the haplotypes reflects this break. In addition, the SAMOVA values match this break since *F_CT_* reaches higher and more significant values in different clusters depending on the gene, but the populations belonging to these two groups were always kept separate: west and southeast. Each genetic marker gave us a higher *F_CT_* value when grouping different populations (Table 2). The 28S gene: *K* = 6 with *F_CT_* = 0.8108 (*p*-value = 0.000); H3 gene: *K* = 9 with *F_CT_* = 0.55096 (*p*-value = 0.00880); H2A gene: K = 2 with *F_CT_* = 0.67267 (*p*-value = 0.09677), Wg gene: *K* = 2 with *F_CT_* = 0.58272 (*p*-value = 0.00098); and COI gene: *K* = 3 with *F_CT_* = 0.81407 (*p*-value = 0.00293). 

#### 3.3.4. Phylogenetic Analysis 

The phylogenetic analyses of the four nuclear genes estimated by Bayesian inference show a clear division between the haplotypes belonging to these two groups (Figure 5). The lnL value was −4785.16 with an effective sample size of 2273. It should be noted that the tree was rooted using the species *Membracis luizae* and the lineages of *M. mexicana* were divided into the populations of the west (in red) and the populations of the southeast (in blue). The phylogenies estimated for each gene also produce the same pattern (see Appendix A). For the COI (Appendix A) and Wg (Appendix A) phylogeny, we were able to include more sequences from the GenBank of the distribution of *M. mexicana* outside of Mexico (two individuals: one from Guatemala and the other from Panama) and one individual for each of the other three species (*M. trimaculata, M. flava,* and *M. foliata).*

It is important to note in Figure 5 that the western lineage corresponds to the Pacific lowlands and the TMVB (including the Zacapu population, whose helmet is different), and the southeast lineage corresponds to the Veracruzan and Yucatan Peninsula provinces. 

#### 3.3.5. Historical Demography 

The historical demographic reconstruction using both data sets, the four nuclear markers, and mitochondrial information showed a recent population expansion within less than 1 Ma (Appendix A). The *Ne* of females increased from 7 to 20, while the nuclear genes show that the increase in *Ne* was from 5 to 80. 

## 4. Discussion

In this study, we explored how the morphological and genetic variation of *M. mexicana* is distributed over a wide geographic scale. Richter [10] mentioned that the different morphs of Membracis species could be explained by the identity of the host plant. In our work, we did not find much evidence for this, so the role of the host plant in morphological variation is still not clear. One limitation of our study is that most of the insects were found and sampled on *Terminalia catappa* and, consequently, our sampling design could not be properly used for testing the Richter hypothesis. Our data suggest that the geographic component partially explains the phenotypic variation in this species. In particular, this is apparent by the analysis of the centroid size of the helmet, since type 1 is larger than types 2 and 3, and this is found in the TMVB province, which suggests that altitude is playing a role in this trait. This pattern has been seen in other insects such as *Heliconius* butterflies, where individuals with larger wings are found at higher altitudes. This pattern is found at both intraspecific and interspecific levels [27]. 

As expected, we found more genetic diversity in the TMVB province compared with the other provinces. Three genes showed haplotypic diversity values above 0.75 in this province (Figure 3). Moreover, this pattern of genetic variation is consistent with two lineages in southeastern and western areas (Figure 5). The western lineage (Pacific Coast and TMVB provinces) corresponds to helmet types 1 and 2, and the southeast lineage (Mexican Gulf and Yucatan Peninsula provinces) corresponds to helmet type 3. The probable cause of this genetic break that distinguishes these two genetic lineages could be the magmatic activity of the Veracruz area during the Pleistocene. This has been described in other insect species like *Canthon cyanellus* (Coleoptera: Scarabaeidae), where the separation between lineages in this area occurred in a time range of 1.63 to 0.91 Ma [28]. We found for *M. mexicana* that the divergence occurred between 2 and 1 Ma. 

It is difficult to compare the phylogeographic pattern of *M. mexicana* with other species because its distribution covers a wide altitudinal range (from 0 to 2240 MASL) while in published work, the studied species are either from highlands or lowlands. For example, species only distributed in biogeographic provinces covering highlands like the *Peromyscus melanophrys* group (Rodentia: Muridae) [29] and *Amazilia cyanocephala* (Apodiformes: Trochilidae; [30]) show genetic barriers explained by lowlands between mountain systems. On the other hand, species only distributed in biogeographic provinces covering lowlands show the mountain systems as barriers to genetic flow. Also, the genetic lineages of Neotropical species have been found predate the closure of the Isthmus of Panama, for example, *Boa constrictor imperator* (Squamata: Boidae), which shows two lineages: one at the Pacific coast and other at the Mexican Gulf that diverged 7 Ma [31], and *Dasypus novemcinctus,* with two lineages that diverged prior to the closing of the Panama Isthmus (Cingulata: Dasypodidae; [32]). Our data indicate that the southeastern and western lineages of *M. mexicana* differentiated in Mexico after the closing of the Isthmus of Panama. It is possible that the southeastern lineage is more ancestral because in the phylogenetic estimates of Wg and COI, we incorporated sequences from the southern distribution of *M. mexicana* (Guatemala and Panama), and the haplotypes are clustered with southeastern lineage (Appendix A); however, we could only incorporate two samples from Central America, which means that more samples from a wider area in the neotropics are needed to test this hypothesis. 

The reconstruction of historical demography using nuclear and mitochondrial genes shows very similar patterns since both Skyline Plots show a constant size of the population for a long time and a recent population expansion. The only difference between mtDNA and ncDNA is that ncDNA allowed us to visualize the dynamics up to 6 million years ago, while mtDNA only shows up to 3.5 million years ago. The mutation rates for mitochondrial and nuclear sequences made it possible to see that the population expansion of these members occurred recently and is temporally consistent. The expansion appears to have started approximately 150,000 years ago and continues to the present. Given the distribution of genetic and morphological variation, we could suggest that *M. mexicana* could be two species of very recent origin, less than 1 Ma, and that the shared haplotypes between these two species are due to ancestral polymorphism. The morphology allows us to distinguish between both groups, but a review of their genitalia and an experimental test are needed to see if there is no longer the possibility of reproducing between these two groups.

In conclusion, the TMVB harbors more intraspecific variations of *M. mexicana* than other biogeographic provinces. The relationship between the morphology and genetics of *M. mexicana* shows that geography plays an important role in population structuring. Finally, we found that the southeastern lineage matches with morphotype 3 and the western lineage with morphotypes 1 and 2.

## Figures and Tables

**Figure 1 insects-14-00704-f001:**
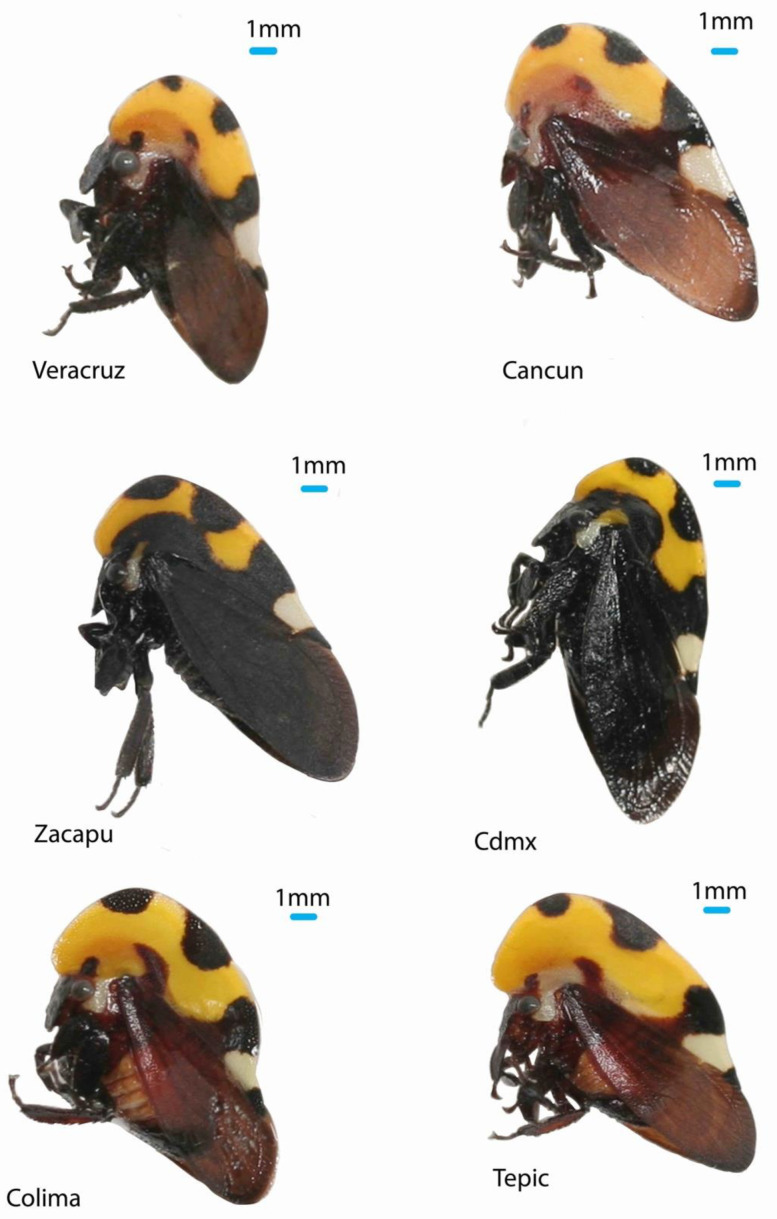
Phenotypic variation of *Membracis mexicana* in Mexico. Cdmx (abbreviation for Mexico city), Veracruz, Cancun, Zacapu, Colima, and Tepic are the localities from which the insects were collected.

**Figure 2 insects-14-00704-f002:**
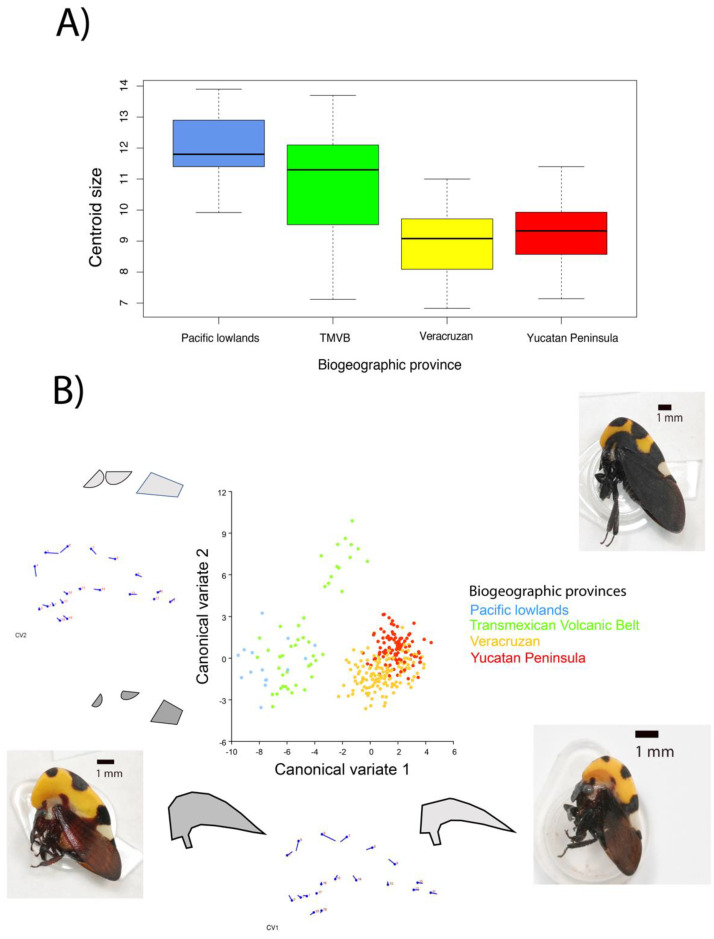
(**A**) Centroid size of the helmet and (**B**) CVA of *Membracis mexicana* by biogeographic province.

**Figure 3 insects-14-00704-f003:**
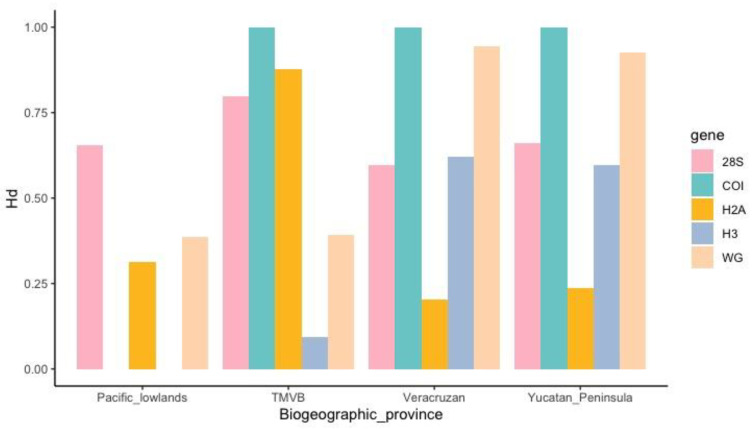
Haplotype diversity (*Hd*) by biogeographic province. In Pacific lowlands, *Hd* of H3 is 0 and we were not able to amplify COI sequences in this region.

**Figure 4 insects-14-00704-f004:**
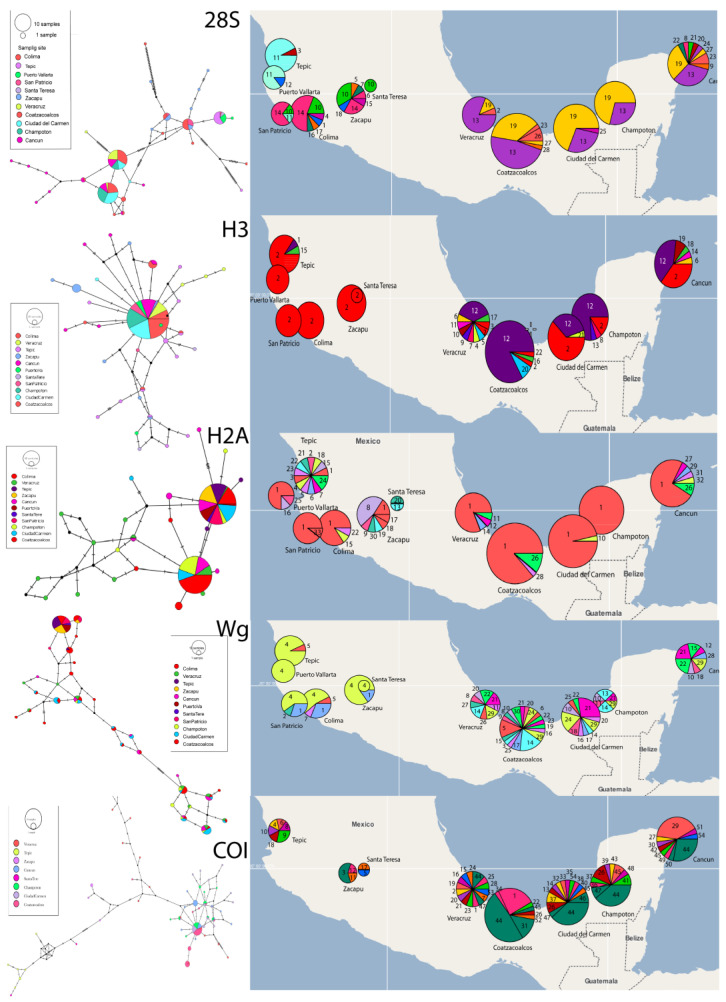
Geographic distribution of genetic variation. Left, gene haplotype networks of genes 28S, H3, H2A, Wg, and COI of *Membracis mexicana*. Circle sizes represent haplotype frequencies, which are colored in each population sampled. Right, distribution and frequency maps of the haplotypes of genes 28S, H3, H2A, Wg, and COI of *Membracis mexicana*. See Appendix A for high-resolution images.

**Figure 5 insects-14-00704-f005:**
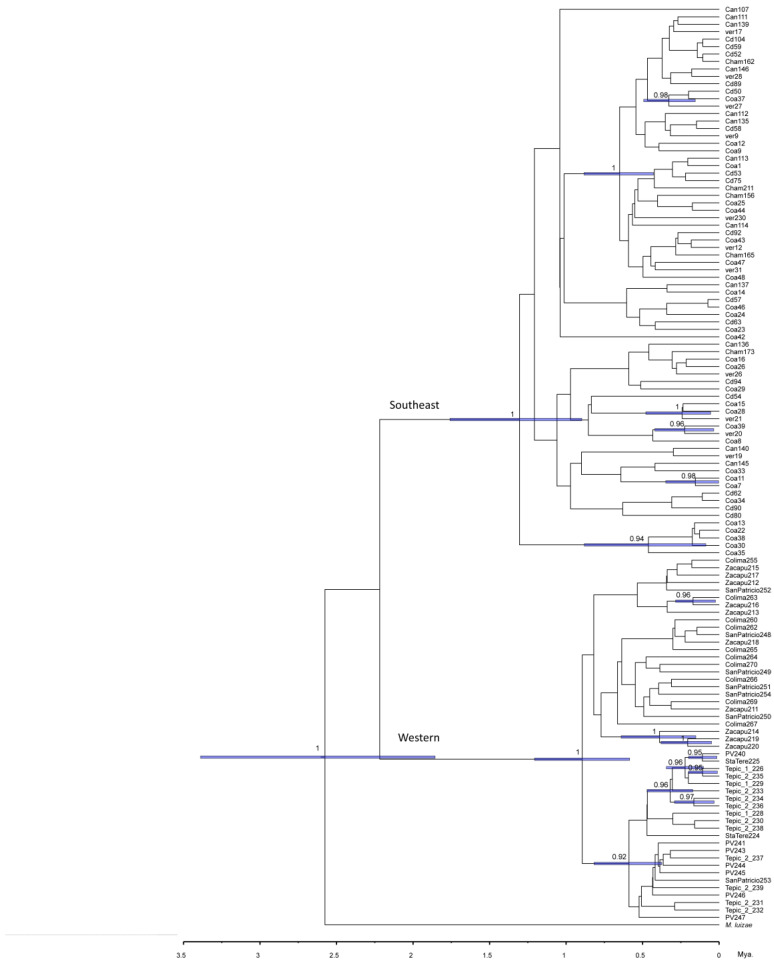
Phylogeny of *Membracis mexicana* using Bayesian inference and four nuclear genes: 28S, H2A, H3, and Wg. Numbers on the branch show the posterior probability. Outgroup *Membracis luizae*.

**Table 1 insects-14-00704-t001:** Sampling localities of *Membracis mexicana* in Mexico. Biogeographic provinces: Yucatan Peninsula (YP), Veracruzan (V), Trans-Mexican Volcanic Belt (TMVB), and Pacific lowlands (PL).

Number	ID	Latitude, Longitude	Host Plant	Sample Size Per Host Plant	State	Country	Sampling Date
1	Cancun (YP)	21°10′08.3″ N 86°51′30.9″ W	*Terminalia catappa, Ficus carica*	18, 8	Quintana Roo	Mexico	December 2019
2	Champoton (YP)	19°21′01.0″ N 90°43′40.3″ W	*Terminalia catappa*	30	Campeche	Mexico	December 2019
3	Ciudad del Carmen (V)	18°39′05.7″ N 91°49′25.1″ W	*Terminalia catappa*	30	Campeche	Mexico	December 2019
4	Coatzacoalcos (V)	18°08′53.1″ N 94°27′02.4″ W	*Terminalia catappa*	30	Veracruz	Mexico	December 2019
5	Veracruz (V)	19°11′13.3″ N 96°07′37.2″ W	*Terminalia catappa*	25	Veracruz	Mexico	May 2019
6	Tepic (TMVB)	21°30′08.2″ N 104°54′09.8″ W	*Spondias purpurea, Terminalia catappa*	4, 11	Nayarit	Mexico	May 2021
7	Puerto Vallarta (PL)	20°38′28.5″ N 105°12′49.0″ W	*Artocarpus altilis, Terminalia catappa*	3, 5	Jalisco	Mexico	May 2021
8	San Patricio (PL)	19°13′25.2″ N 104°42′48.0″ W	*Terminalia catappa*	7	Jalisco	Mexico	May 2021
9	Colima (TMVB)	19°14′10.7″ N 103°44′29.3″ W	*Cassia fistula, Terminalia catappa*	7, 6	Colima	Mexico	May 2021
10	Santa Teresa (TMVB)	19°58′18.7″ N 101°38′05.9″ W	*Heimia salicifolia*	2	Michoacan	Mexico	May 2021
11	Zacapu (TMVB)	19°49′22.9″ N 101°47′17.8″ W	*Salix babylonica*	13	Michoacan	Mexico	May 2021

**Table 2 insects-14-00704-t002:** Results of SAMOVA for each gene sequence showing optimal clustering of *Membracis mexicana* populations.

Gene	Clustering	*K*	*F_CT_*	*p*-Value
28S	(Tepic, Puerto Vallarta) (Colima) (San Patricio) (Santa Teresa) (Zacapu) (Veracruz, Cancun, Champoton, Ciudad del Carmen, Coatzacoalcos)	6	0.8108	0.00000 ± 0.00000
H3	(Puerto Vallarta, Santa Teresa, San Patricio) (Tepic) (Colima) (Zacapu) (Veracruz) (Coatzacoalcos) (Ciudad del Carmen) (Champoton) (Cancun)	9	0.30013	0.00782 ± 0.00242
H2A	(Santa Teresa) (Champoton, Tepic, Zacapu, Ciudad del Carmen, Veracruz, Coatzacoalcos, Puerto Vallarta, Cancun, Colima, San Patricio)	2	0.67267	0.09677 ± 0.00848
WG	(Colima, San Patricio, Zacapu, Puerto Vallarta, Tepic, Santa Teresa) (Coatzacoalcos, Veracruz, Champoton, Ciudad del Carmen, Cancun)	2	0.58272	0.00098 ± 0.00098
COI	(Tepic, Santa Teresa) (Zacapu) (Ciudad del Carmen, Veracruz, Coatzacoalcos, Champoton, Cancun)	3	0.81407	0.00293 ± 0.00164

## Data Availability

The data that support the findings of this study are openly available in GenBank https://www.ncbi.nlm.nih.gov/genbank/ (accessed on 14 June 2023). The ID numbers are OR123974-OR124006 for the H2A partial gene sequence, OR124007-OR124028 for the H3 partial gene sequence, OR124029-OR124058 for the Wg partial gene sequence, OR120258-OR120285 for the 28S partial gene sequence, and OR120103-OR120156 for the COI partial gene sequence.

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
