# Peer review of "Helmet Shape and Phylogeography of the Treehopper *Membracis mexicana"

_insects, 2023, doi:10.3390/insects14080704_

Round 1

Reviewer 1 Report

This paper reports results of a phylogeographic and morphometric study of a widespread neotropical treehopper species, Membracis Mexicana. The authors collected specimens from throughout the range of this species in Mexico as well as incorporating some sequence data from previous studies for specimens collected elsewhere in Central America, and sampled individuals on different host plants. Their morphometric and genetic analyses consistently recovered 2 distinct groups representing southeastern and western region of Mexico and there was some finer-scale geographic structure within these regions. Haplotype diversity was highest in the Transmexican Volcanic Belt, as predicted by the complex geologic history of this region. Overall the study is very well done. This is the first detailed phylogeographic study of a neotropical treehopper and it is important because the species utilizes a variety of host plants and occurs across a broad elevational as well as geographic range. The authors also do a good job of placing the work into the context of other phylogeographic studies that focused on different groups of organisms. I think the paper can be accepted after some minor revisions:

1.     Carefully check the spelling of all scientific names and italicize scientific names consistently.

2.     Explain the abbreviation “Cdmx” in Figure 1.

3.     Discuss whether the southeastern and western populations could represent different species; there seems to be strong phylogenetic and morphological evidence supporting this.

Overall the English is very good but could be improved with some minor editing.

Reviewer 2 Report

Dear authors,

The presented paper has significant value for the comprehensive scientific. The morphological characteristics of the helmet, genetic variations, and geographical distribution of the studied species Membracis mexicana are logical and methodological, very well presented and analyzed.

However, the lacking element in the paper is other measurements of the species. The authors use the terminology for the description size of the helmet as large and small. Nevertheless, the difference between males/females and the total body length can be significant data compared with the helmet size in the presented types 1, 2, and 3.

 Line  219; Membracis mexicana: size, width, and size of the dorsal spots...What are the differences between size and width? It is not apparent; the size can be the length in this context.
